# Laboratory-Scale Preparation and Characterization of Dried Extract of Muirapuama (*Ptychopetalum olacoides* Benth) by Green Analytical Techniques

**DOI:** 10.3390/molecules25051095

**Published:** 2020-02-29

**Authors:** Ester Paulitsch Trindade, Franklin Teixeira Regis, Gabriel Araújo da Silva, Breno Nunes Aguillar, Marcelo Vítor de Paiva Amorim, George Leandro Ramos Ferreira, Cícero Flávio Soares Aragão, Lílian Grace da Silva Solon

**Affiliations:** 1Post-Graduate Program in Pharmaceutical Sciences—PPGCF, Federal University of Amapá—UNIFAP, Rod. Juscelino Kubitschek, Km2, Macapá-AP 68903-419, Brazil; fkregis@hotmail.com (F.T.R.); prof.gabriel.araujo@gmail.com (G.A.d.S.); 2Organic Chemistry and Biochemistry Laboratory, State University of Amapá, UEAP, Avenida Presidente Vargas, 650, Macapá-AP 68900-070, Brazil; 3Drug Quality Control and Bromatology Laboratory—LCqB, Federal University of Amapá—UNIFAP, Rod. JuscelinoKubitschek, Km2, Macapá-AP 68903-419, Brazil; aguillar.farmacia16@gmail.com; 4Research Group on Food and Medicines—NUPLAM, Federal University of Rio Grande do Norte—UFRN, Natal-RN 59072-970, Brazil; celovieee@gmail.com; 5Drug Quality Control Laboratory, Pharmaceutical Sciences Department, Federal University of Rio Grande do Norte-UFRN, Av. General Cordeiro de Farias s/n, Natal-RN 59012-570, Brazil; georgeleandro2@hotmail.com (G.L.R.F.); cicero.aragao@yahoo.com.br (C.F.S.A.)

**Keywords:** *Ptychopetalum olacoides* Benth, spray drying, green analytical techniques, UHPLC, NIR, SEM, dried extract, Analytical Eco-Scale, Muirapuama

## Abstract

This work reports on the preparation of a drying process from the ethanolic extract of Muirapuama and its characterization through green analytical techniques. The spray-drying processes were performed by using ethanolic extract in a ratio of 1:1 extract/excipient and 3^2^ factorial design. The properties of dried powder were investigated in terms of total flavonoid content, moisture content, powder yield, and particle size distribution. An analytical eco-scale was applied to assess the greenness of the developed protocol. Ultra-high performance liquid chromatography (UHPLC)with reduced solvent consumption in the analysis was compared to the conventional HPLC method. A Fourier transform near-infrared (FT-NIR) spectroscopic method was applied based on the principal component scores for the prediction of extract/excipient mixtures and partial least squares regression model for quantitative analysis. NIR spectroscopy is an economic, powerful, and fast methodology for the detection of excipient in muirapuama dried extracts, generating no residue in the analysis. Scanning electron microscopy (SEM) images showed samples with a higher concentration of excipient, presenting better morphological characteristics and a lower moisture absorption rate. An eco-scale score value of 85 was achieved for UHPLC and 100 was achieved for NIR (excellent green analysis). Above all, these methods are rapid and green for the routine analysis of herbal medicines based on dried extracts.

## 1. Introduction

*Ptychopetalum olacoides* Benth is a typical plant from the Amazon rainforest that is used as a tonic among local communities to treat “nerve weaknesses”. It is popularly known as muirapuama, marapuama, and miratã. The traditional use of alcoholic infusions of the roots and ethanolic extracts of *P. olacoides* (EEPO) suggests the presence of bioactive compounds that could present interactions with the dopaminergic system, that is, presenting aphrodisiac, antidepressant, antitremor, and appetite modulatory characteristics, among others [1].

Several muirapuama-based products are easily found in the market in different formulations such as powders, capsules, tinctures, and extracts mixture. For these formulations, different forms of plant preparation are used [1].

It is possible to transform an extractive solution into a powder through dried technological processes. Depending on the type of solvent used to prepare an extractive solution, spray drying may be an alternative, since this technique has presented wide applications in the preparation of pharmaceutical powders with specific characteristics such as particle size and shape [2]. The powder state ensures greater concentration, stability, and ease of standardization of the active principles present in plants, ease of transport, lower risks of microbial contamination, and less space required for product storage, representing advantages that increase the added value of the product [3].

The quality of the herbal medicines formulas depends on the production process employed, the quality of the vegetable raw material, and the excipients used in its production. The importance of plant raw material stems from its nature being definable as a complex mixture of chemical constituents, the levels of which can vary considerably, depending on environmental and/or genetic factors [4,5].

The conduction of many studies involving botanical extracts is becoming important, and they demand analytical methodologies for the identification and characterization of their constituents, requiring standardization assurance, reproducibility, efficacy, and safety in the results obtained [6]. Thus, rigorous analytical control is indispensable at all technological stages of transformation from plant raw material to a final pharmaceutical form.

It is known that aligning the quality in the results obtained from analytical techniques with the environmental standards is a challenge. Since the use of some reagents and solvents can generate toxic waste, finding new techniques and/or refining the existing ones to reduce laboratory wastes or eliminate them has resulted in the emergence of green chemistry [7]. It is known that the impacts of analytical methodologies on the environment and human safety draw attention to the scientific community. In this context, it becomes mandatory to take into account the greenness of the method, such as the use of green solvents, safety of reagents, no need for derivatization, low energy consumption, and low or no waste generation, among others. Therefore, to evaluate these parameters, analytical tools have been used to calculate the green analytical chemistry attributes [8,9]. 

An analytical eco-scale was first proposed by the group of Prof. Jacek Namieśnik [10] through the adaptation of a green organic synthesis evaluation methodology [11]. Since traditional green chemistry metrics usually do not account for the environmental impact, the eco-scale was proposed to encompass all the parameters of green analytical chemistry, such as energy consumption, analyst occupational hazard, solid waste generated, amount and type of solvents, and other chemical reagents. The basis of the scale is expressed as a number that initially has the value of 100 after penalty points are subtracted from each parameter [10]. Due to its versatility and ease of use, the eco-scale has been widely applied to assess the greenness of an analytical methodology.

Based on this trend, companies have invested in the development of analytical devices that meet the principles of green chemistry. For example, through the development of the UHPLC system, many analytical applications have proven the reduced consumption of solvents in addition to the analytical speed and reduced energy consumption per sample when compared to the conventional HPLC [12,13]. Likewise, direct measurement devices have shown advances in the instrumentation, becoming more powerful; for example, the NIR correlates with chemometry to quantify with high selectivity and analytical sensitivity an analyte in the presence of matrix interferents with no use of reagents and no waste being produced [14].

In this context, this work proposes the preparation of muirapuama dried extract by spray-drying (SD) technology employing a factorial design; the evaluation of the particle size and shape of the obtained powder by Scanning Electron Microscopy (SEM); the analysis of the dried extract by ultra-high performance liquid chromatography (UHPLC) and Fourier-transform near infrared Spectroscopy(FT-NIR), in terms of total flavonoids content; and, the greenness assessment of the methodologies by using the analytical eco-scale.

## 2. Results

### 2.1. Spray-Drying Process and Characterization by SEM

The spray-drying process of the extractive solutions was performed in three steps as described: In the first stage, only the EEPO was used (Extract ESB); in the second step, colloidal silicon dioxide (Aerosil^®^) excipient was added to the EEPO in a 1:1 ratio (Extract ESA);and in the third stage, a 2³ factorial design was applied.

The yield for the dried extracts obtained in the first and second steps was calculated considering the initial mass of 2.62 g of extractives. The yields obtained were 36.57% and 38.22%, respectively. The calculation of the percentage of Aerosil^®^ used considered that every 300 mL EEPO was 1.51 g of extractives; therefore, the added amounts of Aerosil^®^ were 0.45 g for samples at 30%, 0.30 g for samples at 20%, and 0.15 g for samples at 10%.

By applying 3^2^ factorial, we obtained nine samples of dried extracts. Two samples (number 08 and 09) were collected only in the cyclone part of the device. The drying yields ranged from 4.7% to 38.22%, and it was possible to observe that the highest yields were for samples with higher concentrations of Aerosil^®^. The amount of the adjuvant influenced the physical behavior of the samples, since those with the highest concentration of Aerosil^®^ were more resistant to the effects of hygroscopicity, which may be related to lower water retention by the dry extract during the drying process, as well as to the physicochemical characteristics provided by this adjuvant [15].

According to the results obtained by Vasconcelos et al. [15] and Loch-Neckel et al. [16] in studies with the dry extracts of *Schinus terebinthiflius raddi* and *Haematococcus pluvialis*, respectively, they showed that the higher concentrations of Aerosil^®^ influenced the drying process yield. However, the variation of the inlet temperature had no significant influence. Tan et al. [17] studied the effect of spray dryer inlet and outlet temperature on the physicochemical properties in the encapsulation of *Momordica charantia* L. dry extract, using maltodextrin and gum arabic as adjuvants. The results showed that the temperature influences the retention of bioactive compounds and the formation of particle films. The ideal inlet temperature found was 140 °C.

#### SEM Analysis

Some chemical and physical properties of drugs are related to the characteristics of the particle size, shape, and morphology. Such characteristics influence the formulation homogeneity, dissolution rate, and bioavailability of bioactive compounds, which are tested for the performance study of the desired intermediate and final products [16].Table 1 shows the averages of the number of particles calculated by the ImageJ^®^ software (ImageJ^®^ 64-bit Java 1.8.0_112 for Windows, Bethesda, MD, USA). Samples 08 and 09 did not result in a sample retained in the collector. Cyclone sample 07 cannot be viewed in the SEM, as it has degraded before analysis due to high hygroscopicity.

SEM images show that there is no uniformity in the particle sizes, as represented in Figure 1. They presented amorphous characteristics, some spherical surfaces, and they are prone to agglomeration. In general, the particles have wrinkled surfaces and irregular shapes, which attribute the low flowability of the powders to these morphological properties [3]. Particle agglomeration can be related to high particle hydration, as shown in the samples with lower Aerosil^®^ concentration in ESB.

Tan et al. [17] state that particles that do not have a uniform structure and cracks have less retention of bioactive compounds. When there is no proper encapsulation of compounds, it can result in greater contact with water (moisture retention) and greater exposure to heat during the drying process. Thereof, such exposures may lead to oxidation and degradation.

The particles presented an average size that ranges between 0.089 and 0.308 µm. These differences in size are related to the morphological irregularities present in the particles and may be associated with the evaporation rates of water and solvent (ethanol) during the spray-drying process [18].Cortés-Rojas, Souza, and Oliveira [19] optimized the production of *B. Pilosa* dried extract by spraying and observed that there was a greater product recovery in the highest proportion of Aerosil^®^, as well as the smallest particle sizes. In their discussion, they attribute the formation of larger particles and agglomerated aspects to the higher residual moisture content due to atomization conditions.

### 2.2. UHPLC Analysis

The calibration curve was obtained in triplicate at concentrations of 2.5 μgmL^−1^ 1.5 µgmL^−1^, 10 µgmL^−1^, 25 µgmL^−1^, 50 µgmL^−1^, 75 µgmL^−1^, and 100 µgmL^−1^ (2% RSD) of quercetin external standard. The flavonoids peaks of the dried extracts were identified by their UV/DAD (diode array detector) spectra [20]. The readings were taken at 340 nm wavelength and the chromatographic run occurred within 30 min. Using the least-squares linear regression, the method’s ability to generate results that were directly proportional to the analyte concentration, which were expressed as a percentage of total flavonoids, could be evaluated. The linear regression equation of the mean curve was y = 15929x − 32226, where y represents the quercetin peak area and x represents its concentration. Linear regression presented a correlation coefficient of R² = 0.9947. 

The overlap of the chromatographic profiles of the obtained dried extracts is presented in Figure 2. Qualitatively, it can be observed that there is no significant change in the chromatographic profile of the extracts to which different proportions of Aerosil^®^ have been added. However, with the addition of this adjuvant, a significant decrease in the peak at 17 minutes of retention time, relative to ESB, was observed.

Total flavonoid content (TFc) was calculated based on the analytical calibration curve for quercetin. The peak areas of the samples were interpolated in their linear regression equation. The total flavonoid content was expressed as µg of EQ (quercetin equivalent) per mg of extract (µgEQ mg^−1^). The area of three peaks at the retention times of 11, 12, and 17 min was used for the calculation, since these peaks presented purity and symmetry for reliable area calculation. The total flavonoids content was 17.81 µgEQ mg^−1^ extract and for the phenolic compounds isolated, the maximum concentration found was 14.48 µgEQ mg^−1^ for ESB and 9.10 µgEQ mg^−1^ for ESA. According to Tian et al. [21], the chemical components of EEPO obtained from the bark of the tree branches in HPLC-DAD indicated a total phenol content, including phenolic acids and flavonoids, of 2.16 mgg^−1^, and for the phenolic compounds isolated, it was 1.04 mgg^−1^.

Figure 3 represents the surface response graph for TFc obtained by 3^2^ factorial design. ANOVA analyses showed that none of the variables is statistically significant, but there is some influence of Aerosil^®^ percentage on the total flavonoid concentration, which can be observed in the most intense red color near the region with the highest concentration of the excipient, presenting a value of *f* = 0.49 and *p* = 0.52 for the input temperature variable, and *f* = 3.15 and *p* = 0.15 for the variation of the Aerosil^®^ percentage. The critical values are those that theoretically would have better results for the maximum flavonoid content in the dry extract, which are inlet temperature values at 148 °C and 23% of Aerosil^®^ in the sample. It is known that the increase in the percentage of Aerosil^®^ in the mixture decreases the concentration of flavonoid in the powdered extract, while a smaller amount of this excipient allows greater hygroscopicity, leading to the instability and degradation of the flavonoids. These data corroborate the study developed by Vasconcelos [15], which investigated the influence of drying temperature and Aerosil^®^ concentration on the characteristics of *Schinus terebinthifolius* Raddi dried extracts.

Comparing to the HPLC conventional analysis reported by Zeirak and collaborators (2010), UHPLC method optimization showed advantages concerning low waste once it could promote a decrease insolvent consumption on the mobile phase. In the method used as a reference [22], the flow rate used was 0.8 mL per min of the mobile phase solvent. In this work, for the analysis of the factorial design, nine samples were analyzed, in triplicate, in a total of 27 chromatographic runs. Starting from this example, 30 min of analysis time could spend 24 mL of solvent by using conventional HPLC and only 6 mL of solvent in the analysis by UHPLC (0.2 mL per min). It may seem a small difference when it comes to the laboratory scale; however, particularly in the pharmaceutical industry, there is interest in reducing the time, the solvent consumption, and therefore the cost of the analysis of large numbers of samples. 

Many authors have discussed the advantages of the use of green methodologies from UHPLC in analytical laboratories [23]. The main advantage was especially a significant reduction in solvent consumptionand consequently the reduction of waste generated by the laboratory. In this case, the optimized method is more economically convenient for routine procedures in the quality control of herbal medicines than traditional methods.

### 2.3. NIR Analysis

For NIR analysis, spectral pre-processing methods were applied to the raw NIR spectral data of all samples to remove unwanted spectral variations due to baseline drifts, light scattering effects, temperature variations, and systematic noise (Figure 4 and Figure 5). 

In this work, NIR analyses propose three objectives: (1) analyze, qualitatively, if the NIR spectra, after mathematical treatment, can differentiate among targets (Aerosil^®^, mixture of extract and Aerosil^®^, and pure dried extract); (2)evaluate, quantitatively, the flavonoid content in mixture of extract and Aerosil^®^ by increasing the absorbance; and (3)assess whether it is possible to differentiate response from the NIR spectra by thedried extract obtained from different parts of equipment (collector or cyclone of the spray dryer).

Then, a calibration model was constructed by principal component analysis (PCA), containing at least 1 sample from each group, to classify the groups by similarity. The samples from each collection site and each mixture of dried extract/Aerosil^®^ are presented as groups in Figure 6.

Based on the PCA, it can be observed that the construction of the model was visually effective, since it was possible to differentiate that pure Aerosil^®^ (green square) and muirapuama dry extract (light blue triangle) are distant from other samples. On the other hand, when we take into account the part of the device, cyclone or collector, the graphic shows that they are quite similar and that the method would only be able to measure the presence or absence of Aerosil^®^ in powder mixture, reiterating the capacity of this method at the level of qualitative analysis.

For the second objective, the construction of the PLS calibration model was applied to insert the leaving-one-out cross-validation method. The values found for root mean square error of calibration (RMSEC), cross-validation RMSEC (RMSECV), and correlation coefficient (R²) were 12.80, 15.06, and 0.018, respectively. Thus, it is possible to verify that there was no correlation between the increase of extract concentration and the NIR response found by Partial Least Squares (PLS) regression. Although this method is not yet able to quantify the presence of flavonoids in the samples, the literature presents some cases of success. Zhao et al. [24] proposed a NIR methodology with chemometric methods for the simultaneous quantification of total flavonoid content and antioxidant activity of *Ginkgo biloba* L. extract, obtaining satisfactory and high potential results. The work developed by Liu Xiaoli [25] pointed out that the NIR technique coupled with multivariate calibration is capable of determining the presence of flavonoids in dried samples of *Flos Sophore Immaturus*. Wulandari et al. [26] studied the use of NIR with chemometrics to determine different levels of flavonoids in leaf extracts of different plants, which pointed out that such methodologies can be used for this purpose. Ali et al. [27] also used coupled methodologies to identify the total polyphenol content in *Acridocarpus orientalis* and Moringa peregrina validating the fast and non-destructive quantification methods for the extracts of the medicinal plants studied.

Thus, further studies are necessary to improve this methodology concerning quantitative analysis.

### 2.4. Analytical Eco-Scale for Greenness Assessment of the UHPLC and NIR Protocols

The proposed UHPLC and NIR methods were evaluated in the context of the green chemistry, showing eco-scale score values, based on penalty points (PPs), of 85 and 100 for UHPLC and NIR, respectively. The perfect green analysis has an eco-scale value of 100 [10]. Table 2 shows the comparison of the analytical procedure parameters and the score obtained for the proposed procedures with the literature protocols for the determination of total flavonoids in herbal preparations. The UHPLC and NIR methods obtained the best results in the context of green chemistry when compared to the protocol described in the literature. 

The eco-scale is very useful for discovering and improving the weakest point in the analytical method. In the case of the UHPLC and NIR proposed protocols, both presented satisfactory eco-scale scores and were classified as “excellent green analysis”. Kalinke et al. [28] presented a methodology for glucose determination using a microfluidic device and adopting a procedure adapted by Gałuszka et al. [10] in order to verify if the proposed device could be classified as a green analytical procedure. The score obtained in this study was 81 points, which represented an excellent green procedure. 

Similarly, El-Shaheny et al. [29] developed methods for the determination of trimebutine and its degradation products and calculated the greenness according to the eco-scale. They found a score of 86 for the conventional fluorimetric method and a score of 85 for the derivative fluorimetric method. These values showed the excellent greenness of the two methods and their suitability as environmentally friendly analytical protocols.

Thus, the main objectives of the green analytical methods were achieved for the UHPLC and NIR methods proposed in this work, demonstrating the reduction of reagents, energy consumption, and wastes. 

## 3. Materials and Methods 

### 3.1. Reagents and Chemicals

Ethyl alcohol (70% *w/w*) was purchased from VIC PHARMA (Vic Pharma^®^ Industriaand Comercio Ltda, Sao Paulo, Brazil), Milli-Q-USA System ultrapure water (Millipore^®^ Corporation, Bedford, MA, USA), Colloidal Silicon Dioxide (SYNTH^®^), UV/HPLC Grade Acetonitrile (JT BAKER^®^), UV/HPLC Grade Formic Acid (JT BAKER^®^).

### 3.2. Herbal Material

The root samples of *P. olacoides* Benth were collected in Porto Grande, Amapá State, Brazil. The plant material was identified by the herbarium of the Federal University of Pará (UFPA) in Pará city, Brazil, under the number 169916 and registered in the National System for the Management of Genetic Heritage and Associated Traditional Knowledge (SisGen) under code AF451C5. The plant material was dried in an oven at 45 ± 1 °C for 72 h, triturated in a slicer, and stored protected from light and moisture to then be subjected to extraction.

### 3.3. Hydroalcoholic Extract Preparation

The ethanolic extract of *P. olacoides* Benth(EEPO) was prepared by adding the root in ethyl alcohol (70%) in a ratio of 1:10 and subjecting it to ultrasound assisted extraction (QUIMIS-Q3350^®^, São Paulo, Brazil) at 20 kHz for a period of 15 min at room temperature.

### 3.4. Determination of Dried Loss

Then, 2 mL of EEPO were weighed in previously dried petri dishes and desiccated for 30 min. After weighing the samples, they were placed in an oven at 105 °C for an initial period of 3 h, removed, and left to desiccate for 30 min and were weighed again. Then, the samples were returned to the oven for an additional 1 h and then removed and cooled in a desiccator until constant weight [30].

### 3.5. Spray-Drying Process and Morphological Examination 

Dry extracts were prepared in a Büchi® B-290 Mini Spray-Dryer (Buchi, Flawil, Switzerland) with a two-fluid nozzle with a 0.5 mm diameter hole. The drying conditions were as follows: under an N2 atmosphere, a sample injection flow of 1 mL.min-1 and air flow of 400 Lh^−1^.Drying of the extractive solutions was performed in three steps:
In the first stage, only the EEPO was used—ESB;In the second step, Colloidal Silicon Dioxide (Aerosil^®^) excipient was added to the EEPO in a 1:1 ratio—ESA;In the third stage, a 3^2^ factorial design was applied. The factor studies and their level were as follows: spray dryer inlet temperature (160, 150, and 140 °C) and percentage of Aerosil^®^ added to the extract (10, 20, and 30% *w/w*). Only in this step, samples were also obtained from the “cyclone” part of the spray dryer in order to compare the morphology and the total flavonoids content in the dried extracts with the part obtained from the “collector”.

Morphological characterization was performed under a Hitachi^®^ TM3030Plus microscope (Tokyo, Japan)). The photomicrographs were taken at 15.0 kV voltage conditions and viewed at 100, 300, 600, and 1000x of magnification. The images obtained were treated with ImageJ^®^ software to evaluate the particle size, shape, and morphology of the dried extracts [31].

### 3.6. UHPLC Analysis

Total flavonoids analysis were carried out by Ultra-High Performance Liquid Chromatography (Shimadzu®, Kyoto, Japan) consisting of the following modules: diode array detector (SPD-M20A), column oven (CTO-20A), auto injector (SIL-20A), degasser mobile phase controller (DGU-20A5), controller system (CBM-20A), quaternary pump (LC-20AT), and LC-Solution data system (software). The data obtained were exported in .txt files and processed in Origin^®^ OriginPro 2019b Trial software (version 9.65, Northampton, MA, USA) for the better visualization and overlapping of chromatograms.

The chromatographic analysis was suited by the methodology developed by Zeirak et al. [22], who quantified the total flavonoids in *Passiflora edulis* fruit pulp by HPLC-DAD. In our work, we transferred this methodology to a UHPLC-DAD system, in a lower scale, as described: the dried extract was reconstituted by diluting 3 mg of each sample into the mobile phase consisting of 0.1% formic acid(Phase A) and acetonitrile (Phase B); then, it was filtered through a 0.22μm pore nylon syringe filter. An Agilent^®^ Poroshell 120/2.7 µm C18 (4.6 × 50mm) column was used as a stationary phase. Analyses were performed at a flow of 0.2 mLmin^−1^, 3 µL of injection volume, column temperature was maintained at 40°C, and scanning was from 200 to 400 nm. Total flavonoids were determined by an external standard method. The calibration curve was constructed with the flavonoid quercetin as the standard, being represented in the Y-axis the peak area, and in the X-axis, the quercetin concentration helped in the obtaining of a linear regression equation. The standard solutions (2.5 –100µg·mL^−1^) were prepared in methanol 80%. Chromatograms were monitored at 340 nm. It calculated the area of all the well-resolved and symmetric peaks identified by the diode array detector (DAD) as being flavonoids.Total flavonoid content was expressed as mgEQ mg^−1^ of extract.

### 3.7. NIR Analysis

A Near Infrared Spectrophotometer (FT-NIR), MPA^®^ model (Bruker^®^, Bremen, Germany), equipped with an integrating sphere, with liquid sample compartment and probe and OPUS^®^ Spectroscopy Software (version 7.0, Bremen, Germany) was used. The method of choice for obtaining the spectra was through the 32-scan background solids probe, 32-scan analysis, and 32-scan resolution in the 12500 to 4000 cm^−1^ wavelength range. Then, the wavelength data (800 to 2500 nm) were converted for data processing. Analyses were performed in triplicate readings for each sample.

The calculations were performed using the PLS-toolbox of the MATLAB version 6.5 (The Math-Works, Natick, MA, USA). The calculated NIR spectra were log 1/R transformed in the first step, followed by the average spectra for each sample. Different pretreatments such as Smoothing Savitzky–Golay (SGS) (7 window points) followed by MSC (multiplicative scatter correction) and first-order derivative Savitzky–Golay (7 window points) were applied on the spectra to minimize undesirable features such as spectral offset, noise, baseline, and scattering [32].

### 3.8. Analytical Eco-Scale Calculation

Analytical eco-scale values for the greenness assessment of the UHPLC and NIR methods were calculated according to Gałuszka et al. [10] as described: analytical eco-scale score = 100—total penalty points. The result of the calculation is ranked on a scale, where the score: >75 represents excellent green analysis,>50 represents acceptable green analysis, and<50 represents inadequate green analysis [10].

### 3.9. Statistical Analysis

Statistical analysis was performed using STATISTICA^®^ software (version 10, StateSoft-Inc., Tulsa, OK, USA, trial version). The ANOVA test was used to evaluate the significance of the influence of independent variables and their interactions [33]. It was established as a significant parameter *p* < 0.05.

## 4. Conclusions

The morphology of the spray-dried of the powder extracts of muirapuama root presented shape with irregularities and amorphous aspect; for that reason, further studies should be done to improve the shape and the flowability of the powders and avoid particle agglomeration. UHPLC analysis was able to quantify the total flavonoids and presented advantages concerning solvent consumption. Near-infrared spectroscopy was applied for analysis of the muirapuama dried extract and was able to perform qualitative analysis. Both analytical methodologies were considered green, presenting excellent green analysis, according to the analytical eco-scale.

## Figures and Tables

**Figure 1 molecules-25-01095-f001:**
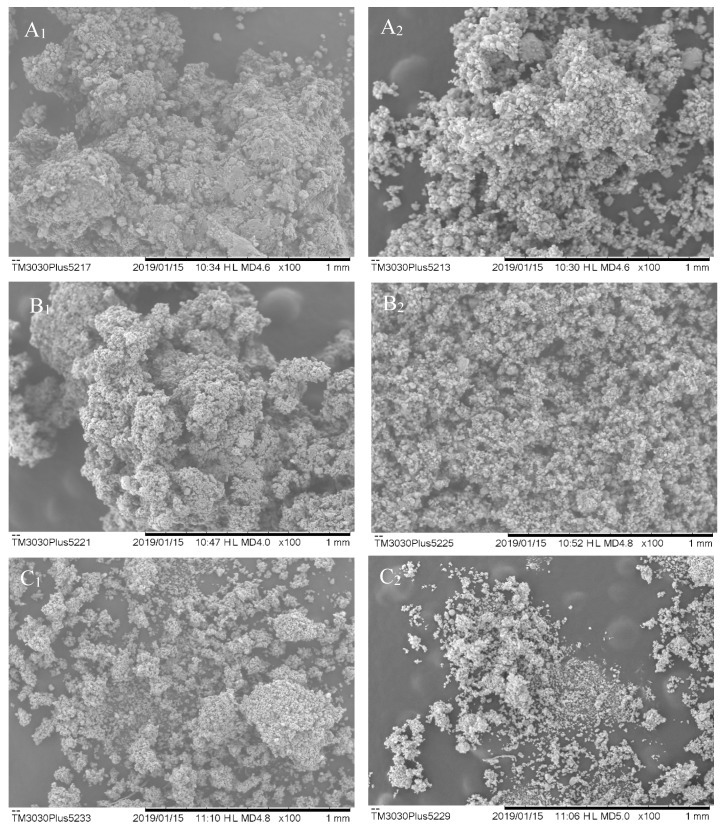
Images obtained by SEM of the dried extracts, which were subscribed as number 1 (indicates the images of samples obtained in the cyclone) and number 2 (those obtained in the collector). (**A**) Sample 01, (**B**) Sample 02, (**C**) Sample 03, (**D**) Sample 04, (**E**) Sample 05, (**F**) Sample 06, (**G**) Sample 07, (**H**) Sample 08, (**I**) Sample 09, (**J**) ESA, and (**K**) ESB *P. olacoides.*

**Figure 2 molecules-25-01095-f002:**
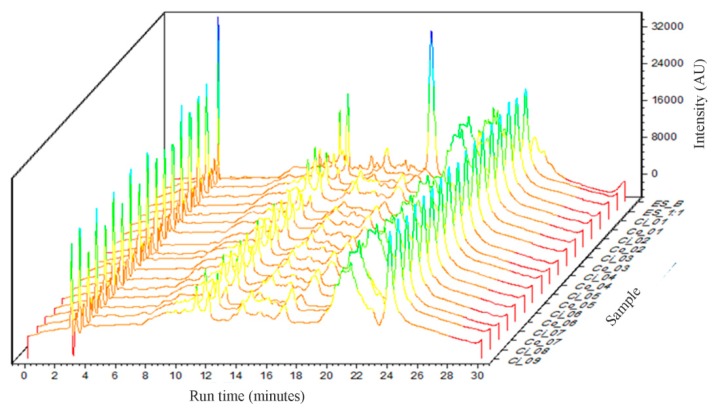
Overlap of chromatographic profiles of the obtained dried extracts.

**Figure 3 molecules-25-01095-f003:**
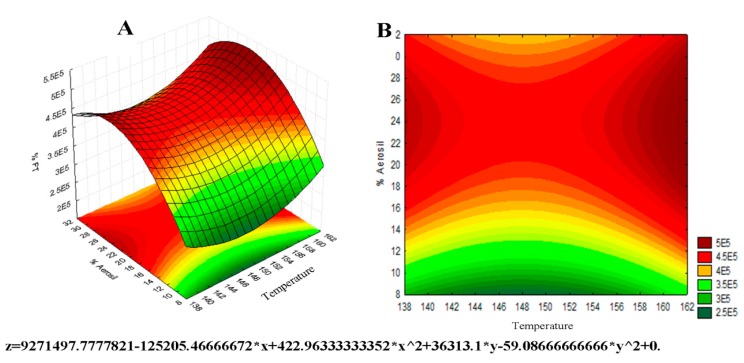
Surface response graph 3^2^ for temperature and Aerosil^®^ percentage factors in relation to total flavonoid content. Surface response in 3D (**A**) and 2D contour plot (**B**).

**Figure 4 molecules-25-01095-f004:**
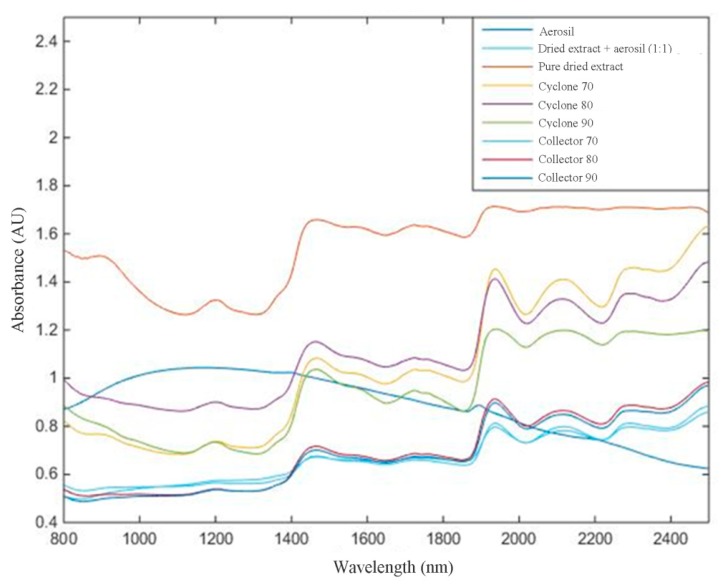
Near-infrared (NIR) spectral data obtained from 800 to 2500 nm with no mathematical treatment.

**Figure 5 molecules-25-01095-f005:**
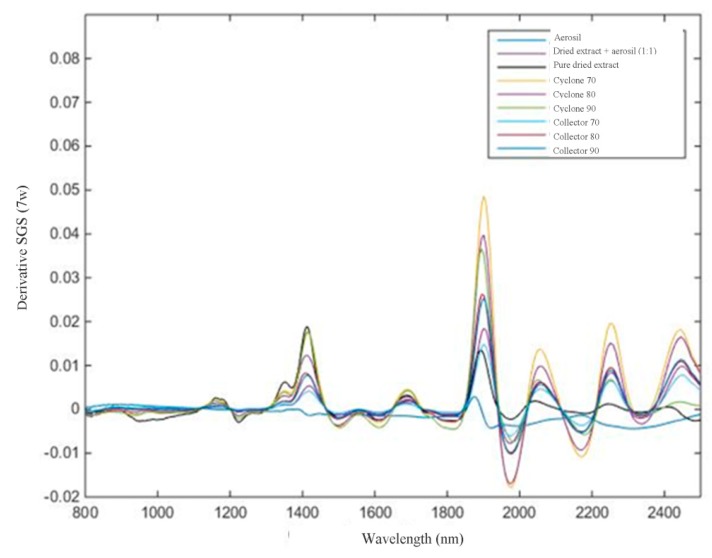
Spectral data obtained from 800 to 2500 nm after Savitzky–GolaySmoothing (SGS) 7w.

**Figure 6 molecules-25-01095-f006:**
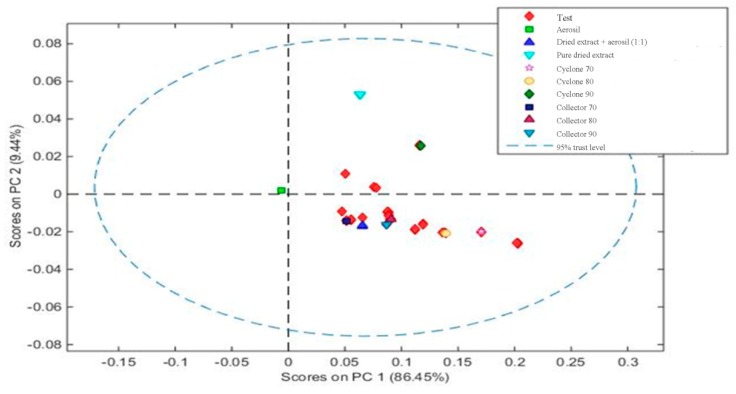
Principal component analysis showing the distribution of the samples based on the mixtures of dried extract/excipient and the extracts obtained in different parts of equipment (collector or cyclone).

**Table 1 molecules-25-01095-t001:** Number of particles (N), average values (x), and standard deviation (σ) of particle areas identified and calculated by ImageJ^®^ software.

	Collector			Cyclone		
Sample	N	$\bar{x}$	σ	N	$\bar{x}$	σ
ESB	8669	0.14	0.16	5254	0.17	0.21
ESA	7148	0.18	0.19	-	-	-
01	4282	0.26	0.23	7432	0.17	0.17
02	5840	0.27	0.21	8196	0.13	0.16
03	11191	0.09	0.11	11965	0.12	0.13
04	9955	0.15	0.16	2381	0.24	0.23
05	8894	0.12	0.16	11513	0.12	0.13
06	13431	0.09	0.13	5136	0.23	0.21
07	4231	0.31	0.24	-	-	-
08	-	-	-	5840	0.23	0.21
09	-	-	-	7903	0.16	0.17

**Table 2 molecules-25-01095-t002:** Analytical parameters comparison and eco-scale score obtained for the UHPLC and NIR methodologies applied to the analysis of herbal medicine dried extract.

Analytical Procedure	Analytes	Reagents Consumed Value [PPs]	Instrumental Hazard Value [PPs]	Eco-Scale Score *
UHPLC-DAD	Flavonoids	Acetonitrile-8	Energy used <0.1 kwh per sample-0	85
		Formic acid-4	Waste-3	
			Occupational hazard-0	

FT-NIR	Flavonoids	Direct measurement	Energy used <0.1 kwh per sample-0	100
	Aerosil^®^	(no reagents consumed)-0	Waste—0	
			Occupational hazard-0	

HPLC-DAD **	Flavonoids	Acetonitrile-8	Energy used ≤1.5 kwh per sample-1	82
		Formic acid-4	Waste-5	
			Occupational hazard-0	

* Analytical eco-scale score = 100-total penalty points (PPs). The result of calculation is ranked on a scale, where the score: >75 represents excellent green analysis; >50 represents acceptable green analysis; and <50 represents inadequate green analysis [10]. ** Based on Zeirak, 2010 [22].

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
