# Peer review of "Laboratory-Scale Preparation and Characterization of Dried Extract of Muirapuama (Ptychopetalum olacoides Benth) by Green Analytical Techniques"

_molecules, 2020, doi:10.3390/molecules25051095_

Round 1

Reviewer 1 Report

Comments:

The authors described a laboratory scale preparation and characterization of dried extract of Muirapuama using green analytical methodology. The title does not explain as to whether the method is industrial scale or laboratory scale. As such, the title should be modified to reflect that the approach is only limited to laboratory scale.  The manuscript needs a thorough linguistic edit.  The manuscript is well written and professionally presented. As such, it can be published after the proposed modifications/edit.  

Author Response

Thank you for the suggestion. The manuscript was fully reviewed and edited by an English-speaking professional and the title was modified to “laboratory scale”.

Reviewer 2 Report

The manuscript submitted by Trindade et al.   “Preparation and Characterization of Dried Extract of Muirapuama (Ptychopetalum olacoides Benth) by a Green Approach”. In a first analysis, the subject of the article is of great interest to the academic community. The article is well written, and it has a good amount of good quality data and hence should be accepted for publication after minor revisions.

The authors should reconsider the number of decimal cases that the results of yields and other calculations present. Did equipment use to allow such big precision?

Please check the legends of the graphs in figure 3, “Temperatura”, is not English.

Besides the optimization (which I agree that multifactorial optimization is a better, more accurate and sustainable approach than one-factor-at a time factor optimization), it would be interesting and will contribute to the scientific community if the influence of each factor is discussed: why is the extraction improved up to a percentage of Aerosil and then starts decreasing?

Above all, these methods are rapid, low cost and green for routine analysis of herbal medicines based on dried extracts.” In the abstract, the authors claim to have low cost and green method to routine analysis. However, the cost of the method was never presented and compared with the traditional ones in the manuscript. This point needs to be discussed in the paper.

Author Response

Thank you for the suggestion. We reconsidered the number of decimal cases that the results of yields and the other calculations presented.

The legends of the graphs in figure 3, “Temperatura” were changed to “Temperature” in the English language.

In factorial design results, the influence of aerosil was discussed and is presented in lines 263-268 of the manuscript.

The cost of the method was discussed concerning solvent consumption and is written on lines 279-286.

Reviewer 3 Report

The manuscript entitled “Preparation and characterization of dried extract of Muirapuama (Ptychopetalum olacoides Benth) by a green approach” propose a preparation of muirapuama dried extract by spray drying (SD) technology using a factorial design. The extract obtained was analyzed using ultra-high performance liquid chromatography (UHPLC) and Fourier-Transform Near Infrared spectroscopy (FT-NIR), in terms of total flavonoids content. In my opinion, this manuscript is suitable to be accepted in Molecules for publication after major revisions.

In my opinion green approach in Title should be removed, in spite of the authors used an analytical Eco-scale, the extract was obtained using an hydroalcoholic solution. Moreover, several works have been done in plant extract using UHPLC, but since the extraction used hydroalcoholic solution the authors avoid green approach.

My concern is related to figures and tables, the authors were not careful in their formatting, since the axes legends are all in Brazilian language and the decimal numbers is point instead of comma. The figure resolution should be improved. The figure captions are not in agreement with results discussion, e.g. Figure 2 in line 216 should be Figure 3.

Figure 3. The authors should pay attention to the significant numbers.

The abstract should be rewritten in order to include background, methods and results.

In the results sections some sentences should be placed in experimental section, e.g. lines 202 to 205.

The authors should add an section of conclusions.

Author Response

Thank you for your observation. The general green approach that was in the title was switched to a green approach only at the level of green analytical chemistry. The experiments were carried out on a laboratory scale using as little extractor solvent as possible in an extraction method considered green (ultrasound-assisted extraction) as described by Tiwari, B.K., 2015; in Ultrasound: A clean, green extraction technology. TrAC Trends in Analytical Chemistry, 71, 100 -109; and thus this method can be considered green when compared to other extraction methods. Although this work had used a green extraction technique, this manuscript is focused on green analytical methods, such as UHPLC and NIR.

The figures and tables have been formatted for the English language and the resolution of the figures was edited to 300 dpi. Changes have been made, thank you. In addition, concerning the equation of the calibration curve, it was added in the experimental section, in topic 3.6.

The conclusion section has been inserted.

Round 2

Reviewer 3 Report

The manuscript entitled “Laboratory Scale Preparation and Characterization of
Dried Extract of Muirapuama (Ptychopetalum olacoides Benth) by a Green
Analytical Techniques” should be accepted in current form.